# Unsupervised 2D Molecule Drug-likeness Prediction based on Knowledge Distillation

## Abstract

With the research significance and application value, drug-likeness prediction aims to accurately screen high-quality drug candidates, and has attracted increasing attention recently. In this regard, dominant studies can be roughly classified into two categories: (1) Supervised drug-likeness prediction based on binary classifiers. To train classifiers, the common practice is to treat real drugs as positive examples and other molecules as negative ones. However, the manual selection of negative samples introduces classification bias into these classifiers. (2) Unsupervised drug-likeness prediction based on SMILES representations, such as an RNN-based language model trained on real drugs. Nevertheless, using SMILES to represent molecules is suboptimal for drug-likeness prediction, which is more relevant to the topological structures of molecules. Besides, the RNN model tends to assign short-SMILES molecules with high scores, regardless of their structures. In this paper, we propose a novel knowledge distillation based unsupervised method, which exploits 2D features of molecules for drug-likeness prediction. The teacher model learns the topology of molecules via two pre-training tasks on a large-scale dataset, and the student model mimic the teacher model on real drugs. In this way, the outputs of these two models will be similar on the drug-like molecules while significantly different on the non-drug-like molecules. To demonstrate the effectiveness of our method, we conduct several groups of experiments on various datasets. Experimental results and in-depth analysis show that our method significantly surpasses all competitive baselines, achieving state-of-the-art performance. Particularly, the prediction bias of SIMILES length is reduced in our method. We will release our code upon the acceptance of our paper.

## 1 Introduction

Predicting the drug-likeness of novel molecules is an essential step during the initial phase of drug discovery. Accurately screening candidates with high likelihood of advancing to clinical trials can significantly reduce the cost of drug development. The drug-likeness of a molecule is associated with its biological efficacy, toxicity, metabolic stability and other properties (Leeson & Springthorpe, 2007; Ursu et al., 2011). As there are so many factors involved, drug-likeness can not be measured by a single quantity simply. Meanwhile, it is impossible to test all molecules in wet-lab experiments. Therefore, how to automatically predict drug-likeness becomes one of the focuses for researchers.

Early researchers summarize some empirical rules to determine whether a molecule has drug potential, such as the Lipinski's rule of five (RO5) (Lipinski, 2004) and the concept of pan-assay interference compounds (PAINS) (Baell & Holloway, 2010), which detects whether a molecule contains structure alerts. Nevertheless, it is observed that there is a certain percentage of approved drugs do not conform to some of these rules (Bickerton et al., 2012a; Yusof & Segall, 2013). Unlike these studies, Bickerton et al. (2012b) introduce the quantitative estimate of drug-likeness (QED) to quantify drug-likeness, so as to avoid the aforementioned strict cut-off. Despite this effort, Beker et al. (2020) show that this metric is indistinguishable for drugs and non-drug molecules.

To overcome the limitations of the above methods relying on human-derived rules, researchers shift their attention to deep learning based models. In this regard, dominant studies (Hu et al., 2018; Beker et al., 2020; Sun et al., 2022) employ real drugs as positive samples and the molecules from other databases as negative samples to train binary classifiers. However, the chemical space of non-

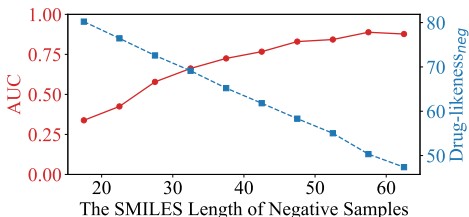

Figure 1: The AUC values of the RNN model (Lee et al., 2022) on ten evaluation sets, each of which contains the same positive samples from FDA-approved molecules and negative samples from ChEMBL dataset with different SMILES lengths.

drug molecules is substantially more expansive than that of drug-like molecules. Consequently, it is impractical to expect a model to cover all non-drug molecules.

In response to this challenge, Lee et al. (2022) resort to unsupervised drug-likeness prediction. They first represent molecules as SMILES and use real drugs to train an RNN-based language model. Subsequently, the drug-likeness of a molecule is calculated based on the likelihood of its corresponding SMILES. Despite of its success, representing molecules as SMILES has some limitations: (1) Utilizing SMILES to represent molecules is less preferable than employing 2D graphs. This is because the drug-likeness of a molecule is more related to its topological structure, which can be naturally represented as a graph. For examples, some substructures with known toxic, mutagenic or eratogenic properties affect the drug-likeness of a molecule, which can not be inferred from SMILES. (2) The drug-likeness score output by the RNN model is influenced by the length of SMILES, regardless of molecular structure. To show this, we use negative samples of different length intervals from ChEMBL and FDA molecules as positive samples to construct 10 test sets and evaluate the performance of the RNN model. As depicted in Figure 1, we can observe that the RNN model tends to assign short-SMILES molecules with relatively high drug-likeness scores.

To tackle these issues, in this paper, we propose a knowledge distillation based unsupervised method for drug-likeness prediction, where two 2D molecular graphs-based models are involved.. Due to representing molecules as 2D graphs, our method not only learn better features but also demonstrates greater robustness to the bias of SMILES length. Specifically, we first collect the small molecules from the ChEMBL database (Mendez et al., 2019) to pre-train a teacher model via two self-supervised tasks: (1) Masked atom modeling (MAM), which is to predict the types of randomly-masked atoms according to unmasked ones and the topological structure of molecules; (2) Masked bond modeling (MBM), which is similar to MAM and aims to predict the types of masked bonds. These tasks are helpful for the teacher model to understand the rich topology information of molecules, which facilitates capturing dominant features for drug-likeness. Subsequently, the knowledge acquired by the teacher model is transferred to a student model which shares the same architecture with the teacher model. During inference, we directly use the gap between the outputs of these two models to measure the drug-likeness of the input molecule.

The reason behind the above scoring lies in the differences in the training data and methods of the two models. Concretely, the teacher model is pre-trained on various molecules, while the student model mimics the output of the teacher model only on real drugs. Therefore, when encountering the input molecule that differs significantly from drugs, the student model have difficulty in mimicking the output of the teacher model, resulting in a gap between the outputs of the two models.

To summarize, the main contributions of our work are three-fold:

- We in-depth analyze the defects of conventional studies on drug-likeness prediction. Particularly, we propose that 2D molecular structures are more suitable for drug-likeness prediction than SMILES. To the best of our knowledge, our work is the first attempt to explore unsupervised drug-likeness prediction using 2D molecular graphs.
- We propose a novel unsupervised method based on knowledge distillation for drug-likeness prediction. Using our method, we train a teacher model and a student model via different datasets and methods, and use the gap between their outputs to quantify the drug-likeness of the input molecule.
- To demonstrate the effectiveness of our proposed method, we conduct several groups of experiments on distinguishing drugs and various types of non-drug molecules. Experimen-

tal results show that our method outperforms all the baselines (Lee et al., 2022; Ma et al., 2022; Niu et al., 2023).

## 2 RELATED WORK

**Handcrafted rules-based drug-likeness prediction.** Early Drug-likeness prediction mainly relies on property-based rules (Lipinski, 2004; Ghose et al., 1999; Oprea, 2000; Zheng et al., 2005), such as RO5 (Lipinski, 2004), which sets thresholds on physicochemical properties. Besides, drug-likeness can also be screened by the representative structural patterns of drugs (Wang & Ramnarayan, 1999; Xu & Stevenson, 2000; Muegge et al., 2001; Ursu & Oprea, 2010). Compared with these binary discrimination methods, QED (Bickerton et al., 2012b) offers a continuous measurement, considering eight molecular properties. However, their main defect lies in the over-restriction to the real drugs or drug candidates, thus may potentially screen out novel drug scaffolds (Yusof & Segall, 2013; Lee et al., 2022).

**Supervised drug-likeness prediction.** To overcome the limitations of handcrafted rules, researchers explore deep learning based models to learn features related to drug-likeness in a data-driven way. Due to the lack of molecules with drug-likeness scores as supervision signals to train a regression model, most works develop binary classifiers, where real drugs are used as positive samples and other molecules such as those from the ZINC database (Sterling & Irwin, 2015) are selected as negative samples. Hu et al. (2018) pre-train an autoencoder on a large scale of molecules, and then develop an autoencoder-based classifier to conduct drug-like/non-drug-like classification. Hooshmand et al. (2021) use a deep belief network for drug-likeness prediction, with every two consecutive hidden layers forming a restricted Boltzmann machine.

Recent studies (Beker et al., 2020; Sun et al., 2022; Cai et al., 2022; Gu et al., 2024) focus on exploiting GNNs for drug-likeness prediction in an end-to-end way. For example, Beker et al. (2020) combines the previous binary classifiers with the Bayesian deep neural network. Sun et al. (2022) improves graph convolutional network with an attention mechanism, achieving better performance. Cai et al. (2022) develops three individual GNN models to evaluate the potential of reaching in vivo, investigational, and approved stages progressively from in-stock compounds. Recently, Gu et al. (2024) introduces more features to predict drug-likeness of molecules.

**Unsupervised drug-likeness prediction.** The above supervised binary classifiers learn features in a data-driven way, so the decision boundaries they learned are inevitably affected by the negative examples selected manually in the training set. To deal with this issue, Lee et al. (2022) first explore an unsupervised drug-likeness prediction model, and adopt generative self-supervised learning to train a RNN model on the SMILES of molecules. They perform pre-training on 10 million molecules from the PubChem database (Kim et al., 2020), and then fine-tune the model only with real drugs to fit their distribution. The drug-likeness score is calculated as the sum of the log values of each token-level conditional probability output by the RNN model. However, they do not directly take advantage of the topological information of molecule graph, resulting in the drug-likeness being affected by the length of SMILES regardless of the molecule structure.

Unlike the above-mentioned studies, our work is the first attempt to explore unsupervised drug-likeness prediction based on 2D molecules. Particularly, inspired by the studies on unsupervised graph-level anomaly detection (Ma et al., 2022; Qiu et al., 2022; Niu et al., 2023; Liu et al., 2023), we propose a knowledge distillation-based unsupervised method for drug-likeness prediction. Using this method, we employ different datasets and training strategies to construct two models, and utilize their representation gap in topological structure to quantify the drug-likeness of the input molecule.

## 3 BACKGROUND

As mentioned above, our method involves a teacher model and a student model sharing the same architecture. Inspired by the state-of-the-art(SOTA) molecule representation models (Zhou et al., 2023; Lu et al., 2023a; Luo et al., 2023), we utilize a two-track Transformer-based molecule encoder as our backbone.

We first represent each input molecule as a graph with atoms as nodes and bonds as edges and then stack Transformer layers to learn two kinds of representations: (1) atom representations $\mathbf{X} \in \mathbb{R}^{N \times d_a}$, where $N$ is the number of atoms and $d_a$ is the dimension of atom representation and (2) atom pair

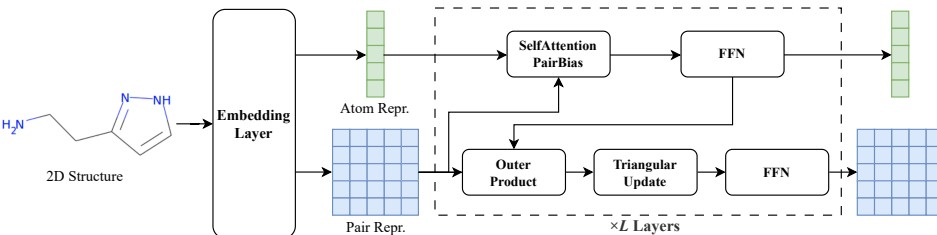

Figure 2: The architecture of our backbone, which contains two tracks to learn atom representations and atom pair representations, respectively.

representations $\mathbf{P} \in \mathbb{R}^{N \times N \times d_P}$, modeling the potential interaction between any two atoms, where $d_p$ is the dimension of atom pair representation. In the following, we give a detailed description to the basic architecture of our backbone. Please note that any 2D molecule representation model is applicable to our unsupervised method.

In the specific implementation, we first introduce an embedding layer to initialize the representations of atoms and atom pairs. Specifically, given atom representations $\mathbf{X} = \{\boldsymbol{x}_i\}_{i=1}^{N}$, we define the initial representation $\boldsymbol{x}_i^{(0)}$ of the $i$-th atom as the sum of its type embedding, degree embedding, and other feature embeddings. More details of features for atom and edge are provided in Appendix A. The atom pair representations $\mathbf{P} = \{\boldsymbol{p}_{ij}\}_{1 \leq i,j \leq N}$ are initialized by the pair-wise position encoding (Ying et al., 2021; Zhou et al., 2023; Lu et al., 2023a). Concretely, the atom pair representation $\boldsymbol{p}_{ij}$ of the $i$-th and $j$-th atoms is initialized by both the shortest path between the $i$-th and $j$-th atoms in the molecular graph and the $k$-th of $n_p$ predefined edge features, such as the edge type. Then, on the top of the embedding layer, we stack $L$ Transformer layers on the top of the embedding layer to encode the above-mentioned molecule graph. At each layer $l$, we sequentially apply several functions to update atom and atom pair representations. Specifically, the atom representation $\boldsymbol{x}_i^{(l-1)}$ is firstly updated by an $\mathrm{Attention}(\cdot)$ function with a bias term to obtain $\boldsymbol{x}_i^{(l)}$. Subsequently, we perform the $\mathrm{OuterProduct}(\cdot)$ function on the updated atom representation $\boldsymbol{x}_i^{(l)}$ and add it to the atom pair representation $\mathbf{P}^{(l-1)}$, enhancing atom to atom pair communication. Finally, $\mathbf{P}^{(l-1)}$ is updated through a $\mathrm{TriangularUpdate}(\cdot)$ function followed by a feed-forward layer to obtain the representation $\mathbf{P}^{(l)}$. Please notice that the effectiveness of these functions has been demonstrated by previous studies (Jumper et al., 2021; Zhou et al., 2023; Lu et al., 2023a). More details and formulas are provided in Appendix B.

# 4 OUR METHOD

In this section, we propose a novel method for drug-likeness prediction. As shown in Figure 3, our method mainly involves two stages to train two models with the same architecture: 1) pre-training stage. At this stage, we directly use a large scale of 2D molecules to pre-train a teacher model, where two pre-training tasks are introduced to learn the knowledge of topological structure; 2) The knowledge distillation stage. Using only real drugs, we then distill the knowledge of the teacher model to the student model, where the outputs of the teacher model are used as supervisory signals. With the above trained teacher and student models, during inference, we directly quantify the drug-likeness of each input molecule as the gap between the atom representations of the above-mentioned models. In the following, we give detailed descriptions of our method.

## 4.1 PRE-TRAINING THE TEACHER MODEL

As mentioned above, we first introduce two tasks to pre-train a teacher model. As analyzed in the previous study (Reiss et al., 2023), the highly expressive representations learned by SOTA pre-training models (Rong et al., 2020; Wang et al., 2022) often fail to detect many simple properties of interest, which reveals that the design of pre-training tasks should be relevant to the specific property of the downstream task. To this end, we carefully design two pre-training tasks, including masked atom modeling and masked bond modeling, so as to learn the topological structures of molecules which are related to the drug-likeness of molecules. Thus, the whole pre-training objective can be

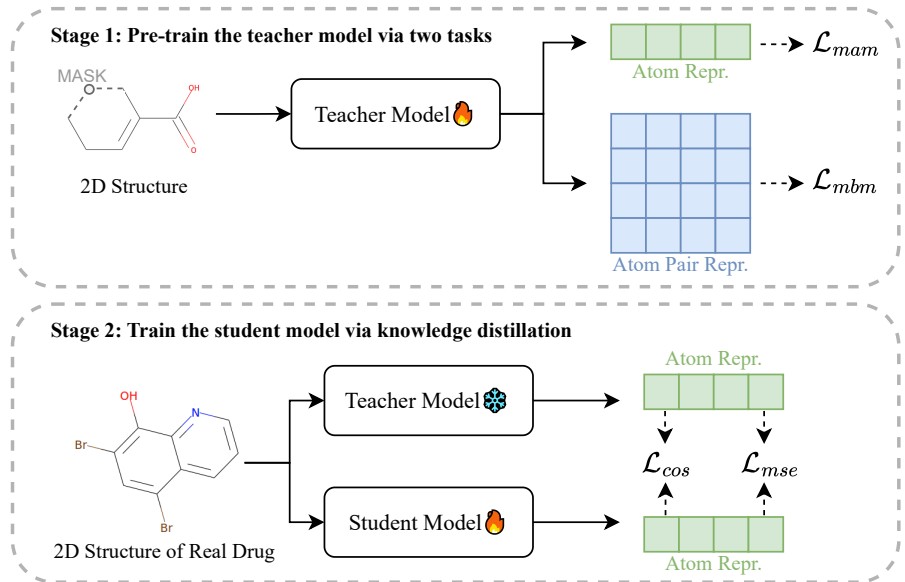

Figure 3: The overview of our method, where the teacher and student models share the same architecture. In the pre-training stage, we train the teacher model on a large scale of molecules, where two 2D pre-training tasks are involved: masked atom modeling and masked bond modeling. At the knowledge distillation stage, we use only real drugs to train student model, which mimics to mimic the representations learned by the teacher model.

decomposed into the following two parts:

$$\mathcal{L}_{pre} = \mathcal{L}_{mam} + \lambda_1 \mathcal{L}_{mam}, \tag{1}$$

where $\mathcal{L}_{mam}$ and $\mathcal{L}_{mam}$ denote masked atom modeling loss and masked bond modeling loss, respectively, and $\lambda_1$ is a tuning parameter that controls the effects of two losses. We detail the two pre-training tasks in the following.

**Task 1: Masked Atom Modeling.** This task is adapted from the concept of masked language modeling in NLP, and has been widely used in molecule representation learning (Hu et al., 2020; Li et al., 2021; Zhou et al., 2023). Specifically, we first represent the atoms with discrete tokens from the dictionary of RDKit. Then, we randomly mask a certain percentage of atoms in the molecule by replacing their atom features with special [MASK] tokens. Subsequently, we introduce an atom prediction head based on the top-layer representations to predict the type of each masked atom. Formally, the training loss $\mathcal{L}_{mam}$ of this task is defined as the cross entropy between the distribution of predicted atom type and the ground-truth type of each masked atom $a_i$:

$$\mathcal{L}_{mam} = -\sum_{\mathcal{G} \in \mathcal{D}} \log \prod_{i \in \mathcal{M}} p\left(a_i \mid \mathcal{G}^M\right), \tag{2}$$

where $\mathcal{M}$ denotes the index set of masked atoms in the masked molecule graph $\mathcal{G}^M$, and $\mathcal{D}$ denotes the pre-training dataset.

Through this task, we can train the model to infer the missing atom information based on the contextual information provided by the surrounding atoms and the molecule graph structure.

**Task 2: Masked Bond Modeling.** We further introduce Masked Bond Modeling (MBM), which is similar to MAM, so as to learn the edge information of molecule graphs. Concretely, we randomly mask a portion of bonds, and then equip the model with a bond prediction head to infer the type of each masked bond according to atom pair representations. Likewise, the training loss $\mathcal{L}_{mbm}$ of this task is also a cross-entropy loss:

$$\mathcal{L}_{mbm} = -\sum_{\mathcal{G} \in \mathcal{D}} \log \prod_{i \in \mathcal{M}} p\left(b_i \mid \mathcal{G}^M\right), \tag{3}$$

where $b_i$ is the $i$-th masked bond.

## 4.2 TRAINING THE STUDENT MODEL VIA KNOWLEDGE DISTILLATION

We then use only real drugs to train the student model, where the outputs of the teacher model are used as supervisory signals. During this process, we freeze the parameters of the teacher model, and train the student model by minimizing the difference between the top-layer outputs of the teacher and student models.

Formally, we define the training objective of the student model as follows:

$$\mathcal{L}_{kd} = \mathcal{L}_{mse} + \mathcal{L}_{cos}, \tag{4}$$

where $\mathcal{L}_{mse}$ denotes the Euclidean distance between representations learned by the teacher and student models, and $\mathcal{L}_{cos}$ means the cosine distance between them. The basic reason behind using two distance functions is that these two distances measure the similarity between the outputs of the teacher and student models from different perspectives, i.e., cosine distance represents directional similarity and Euclidean distance represents the similarity in magnitude. In this way, we can guide the student model more effectively to mimic the representation learning of the teacher model.

More specifically, $\mathcal{L}_{mse}$ is defined as the L2 norm between the top-layer outputs of the teacher and student models, measuring the direct discrepancy between two models:

$$\mathcal{L}_{mse} = \frac{1}{|B|} \sum_{\mathbf{x} \in B} \left( \frac{1}{N} \sum_{i=1}^{N} \left\| \mathbf{x}_i^T - \mathbf{x}_i^S \right\|^2 \right), \tag{5}$$

where $\mathbf{x}_i^T$ and $\mathbf{x}_i^S$ represent the $i$-th atom representations of the teacher and student models, respectively, $|B|$ is the total number of samples in each batch $B$, and $N$ is the number of atoms in a molecule.

Meanwhile, to ensure that the student model aligns in the direction with the teacher model in the feature space, we formulate $\mathcal{L}_{cos}$ as

$$\mathcal{L}_{cos} = \frac{1}{|B|} \sum_{\mathbf{x} \in B} \left( \frac{1}{N} \sum_{i=1}^{N} \left( 1 - cos\left( \mathbf{x}_i^T, \mathbf{x}_i^S \right) \right) \right). \tag{6}$$

## 4.3 DRUG-LIKENESS SCORING

During inference, we feed each input molecule into the teacher and student models to obtain its atom representations, and then measure its drug-likeness in the following way:

$$s = \frac{1}{N} \sum_{i=1}^{N} \left( \| \mathbf{x}_i^T - \mathbf{x}_i^S \|^2 + \left( 1 - cos\left( \mathbf{x}_i^T, \mathbf{x}_i^S \right) \right) \right). \tag{7}$$

Here, we explain the basic intuition behind our function for drug-likeness scoring. **Since the teacher model is trained on large-scale molecules, including non-drug-like and drug-like molecules while the student model is trained only on real drugs, the outputs of teacher and student models will be very similar when the input molecule is drug-like. However, the outputs are not guaranteed to be similar on non-drug-like molecules. Therefore, the representation gap between their outputs can serve as a means to quantify the drug-likeness of each input molecule.**

## 5 EXPERIMENTS

### 5.1 SETTINGS

**Dataset.** We use ChEMBL (Mendez et al., 2019) as the pre-training data for the teacher model, while we adopt the train and test data for the student model following Lee et al. (2022)[1]. In addition, we propose a new test set called BondError. For more details about the datesets, please refer to Appendix C.

---

[1]We also try lower similarity thresholds to extract training data for the student model, and then reconduct experiments. Results reported in Appdendix D validate the generalizability of our method .

**Implementation and Evaluation.** Both teacher and student models consist of 12 layers. During pre-training, we assign the hyper-parameter $\lambda_1$ of $\mathcal{L}_{mbm}$ with 20 (See Equation 1), which is determined on the validation set. And we train the teacher model for 150K steps with 1,000 warmup steps on the pre-training dataset with the weight decay parameter of 1e-4 and a batch size of 128, where atoms are randomly masked at 30%, while bonds at 15%. Besides, we utilize the Adam optimizer with $\beta$ parameters set to (0.9, 0.99) and a learning rate of 1e-4. During distillation, the student model shares the same settings of optimizer, weight decay and batch size with the teacher model, while it is only trained for 3K steps. During this process, the warmup step is reduced to 100, and the learning rate is 5e-4.

Following Lee et al. (2022), we report the area under the receiver operating characteristic curve (AUC). Each group of main experiment is independently run for five times with different seeds, and the mean and standard derivation are reported.

**Baselines.** We choose two kinds of representative drug-likeness prediction methods for comparison: (1) unsupervised methods: QED and RNN; (2) supervised binary classification method: GCN.

- QED (Bickerton et al., 2012b): It is derived from a multi-parameter optimization framework that captures the underlying distributions of molecule properties such as molecule weight, lipophilicity, and hydrogen bond donors and acceptors in real drugs. It can provide a quantified score.

- RNN (Lee et al., 2022): This model stands as the sole of unsupervised drug-likeness prediction model. It is a language model trained on the SMILES of real drugs, using the sum of the log values of each token-level conditional probability to represent drug-likeness.

- GCN (Lee et al., 2022): It is trained with a binary cross-entropy loss function, with World-drug and ZINC15 as the positive and negative sets, respectively.

Given that both our task and unsupervised molecule anomaly detection focus on binary graph classification where only positive samples are available, we also select the following two commonly-used unsupervised molecule anomaly detection models:

- GlocalKD (Ma et al., 2022): Aligned with its method, we randomly initialize and then freeze the parameters of a teacher model. Then, we train a student model by performing knowledge distillation on both node-level and graph-level features. Finally, we calculate the drug-likeness as the gap between the teacher and student models from these two levels.

- HimNet (Niu et al., 2023): It is a graph-based autoencoder, augmented with hierarchical memory modules. The node-level and graph-level memory modules store the patterns of normal samples, restricting the model generalizing to non-drug-like molecules. The graph reconstruction error and the graph approximation error are the drug-likeness score.

## 5.2 MAIN RESULTS

Table 1 shows the main results on different test sets. **Overall, our method consistently exhibits better performance than all unsupervised baselines on four test sets, demonstrating the effectiveness and generalizability of our method.** Furthermore, we draw the following conclusions:

First, QED struggles to differentiate between two categories of molecules on all datasets, limiting its utility in practical applications, which echoes the previous study (Beker et al., 2020). Second, both GlocalKD and HimNet are even worse than rule-based QED on some test sets. This indicates that the naive adaptation of anomaly detection models for drug-likeness prediction, a task involving complex molecular properties, is insufficient in achieving optimal performance. Conversely, our pre-training tasks focusing on molecular structures can effectively capture the features related to drug-likeness well, thereby enabling our method to yield superior results. Particularly, on BondError, our method also achieves the best result, demonstrating that our method is also applicable to the drug-likeness prediction of molecules generated from machine learning based models.

Compared with the supervised baseline, all unsupervised methods perform worse than the GCN model on FDA/ZINC15, which shows an almost perfect AUC value close to 1 (0.991). This is because the GCN model is trained using ZINC15 molecules as negative samples, which, however,

Table 1: The AUC values on the four test sets, where FDA/* means the instances in FDA and * are used as positive and negative instances, respectively. The best result on each test set is marked in bold. $^\dagger$ indicates previously reported scores.

| Model | FDA/GDB17 | FDA/ZINC15 | FDA/ChEMBL | FDA/BondError |
|---|---|---|---|---|
| QED (Bickerton et al., 2012b) | $0.539\pm0.024^\dagger$ | $0.326\pm0.003^\dagger$ | $0.549\pm0.004^\dagger$ | $0.449\pm0.000$ |
| RNN (Lee et al., 2022) | $0.979\pm0.005^\dagger$ | $0.921\pm0.001^\dagger$ | $0.824\pm0.010^\dagger$ | $0.920\pm0.002$ |
| GCN (Lee et al., 2022) | $0.747\pm0.002^\dagger$ | $\mathbf{0.991}\pm0.000^\dagger$ | $0.701\pm0.012^\dagger$ | $0.709\pm0.061$ |
| GlocalKD (Ma et al., 2022) | $0.348\pm0.028$ | $0.579\pm0.008$ | $0.621\pm0.018$ | $0.576\pm0.016$ |
| HimNet (Niu et al., 2023) | $0.581\pm0.139$ | $0.311\pm0.041$ | $0.508\pm0.081$ | $0.419\pm0.066$ |
| Ours | $\mathbf{0.987}\pm0.001$ | $0.950\pm0.001$ | $\mathbf{0.846}\pm0.001$ | $\mathbf{0.927}\pm0.001$ |

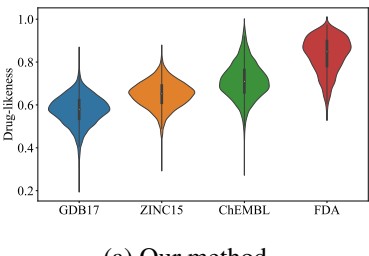

(a) Our method

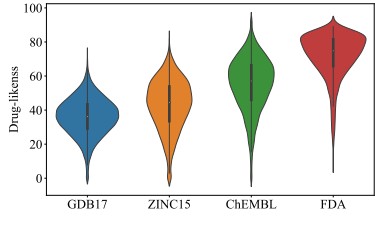

(b) RNN model

Figure 4: The violin plots of drug-likeness output by the RNN model and ours on various datasets.

limits its generalization to other kinds of negative samples, so the GCN model shows lower performance on other test sets.

## 5.3 SCORING FEASIBILITY

Following Lee et al. (2022), we investigate whether our method, which learns solely from real drug data, can naturally quantify the differences in drug-likeness among different types of molecules. To this end, we analyze the distributions of drug-likeness scores output by the RNN model and our method on the FDA-approved molecules (FDA) and three kinds of negative molecules: (1) GDB17: it is a set of highly non-drug-like molecules generated by a graph enumeration method; (2) ZINC15: it represents a chemically accessible space, where molecules are expected to be more drug-potential than those of GDB17; (3) ChEMBL: the molecules in this dataset are with a pChEMBL value of 5.85 or higher, representing a bioactive space and thus expected to be more drug-potential than those of ZINC15.

From Figure 4, we observe that the drug-likeness distributions generated by our method are more concentrated compared to those output by the RNN model, illustrating the effectiveness of our method in distinguishing different kinds of molecules. Furthermore, our average drug-likeness score for each dataset exhibits a gradual increase from the lowest value in GDB17 to the highest value in FDA. This trend aligns with our expectation based on the characteristics of the above-mentioned datasets, supporting the use of our method as a practical tool to quantify drug-likeness. Additionally, our method always rates the molecules in FDA with higher drug-likeness scores, while the RNN model assigns lower scores in some cases. We attribute this issue to the influence of SMILES length on the RNN model, and we will conduct a detailed analysis in the next subsection.

## 5.4 RNN VS. OURS: THE BIAS OF SMILES LENGTH

As described above, the RNN model tends to assign short SMILES with relatively high scores and can not distinguish from drug-like and non-drug-like molecules with short SMILES effectively. By contrast, our model leverages molecular graphs, and thus can alleviate the prediction bias caused by SMILES length. To verify this, we select FDA/ZINC15 and FDA/ChEMBL, both of which contain abundant molecules with different SMILES lengths, and divide each test set into 10 subsets according to SMILES length of its negative samples. Then, we investigate the performance of RNN

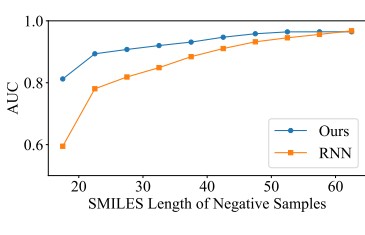
(a) FDA/ZINC15 subsets

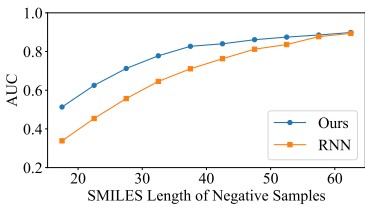
(b) FDA/ChEMBL subsets

Figure 5: The AUC values of the RNN model and ours on subsets divided by the length of SMILES.

Table 2: Results of the ablation experiment.

| Method | FDA/GDB17 | FDA/ZINC15 | FDA/ChEMBL | FDA/BondError |
|---|---|---|---|---|
| Ours | 0.987 | **0.950** | **0.846** | **0.927** |
| *w/o* $\mathcal{L}_{mam}$ | 0.209 | 0.492 | 0.711 | 0.651 |
| *w/o* $\mathcal{L}_{mbm}$ | **0.993** | 0.927 | 0.839 | 0.923 |
| *w/o* $\mathcal{L}_{pre}$ | 0.595 | 0.479 | 0.649 | 0.627 |
| *w/o* $\mathcal{L}_{mse}$ | 0.985 | 0.948 | **0.846** | 0.926 |
| *w/o* $\mathcal{L}_{cos}$ | 0.984 | 0.948 | 0.845 | 0.926 |

and our method on different subsets. Here, in each subset, we still use the same FDA-approved molecules as positive samples.

The AUC values of RNN and our method on each subset are illustrated in Figure 5. As expected, our method exhibits relatively superior performance compared to RNN on most subsets. In contrast, the AUC value of RNN increases with the SMILES length of negative samples, and RNN fails to differentiate between negative samples with short SMILES length and positive samples. For instance, for the negative samples with SMLES length ranging from 15 to 25, the AUC values of RNN are less than 0.5, which are even worse than those of the random classification. In conclusion, these results demonstrate the effectiveness of our method in mitigating the prediction bias of SMILES length.

## 5.5 ABLATION STUDIES

We conduct more experiments to investigate the effectiveness of different components in our method.

**The effect of pre-training tasks.** In this group of experiments, we remove $\mathcal{L}_{mam}$ and $\mathcal{L}_{mbm}$ to obtain two variants, denoted by *w/o* $\mathcal{L}_{mam}$ and *w/o* $\mathcal{L}_{mbm}$. Besides, inspired by Ma et al. (2022), we consider a variant *w/o* $\mathcal{L}_{pre}$, which uses a randomly-initialized teacher.

Table 2 report the performance of our method and variants. From line 2, we can observe an obvious performance decline when $\mathcal{L}_{mam}$ is removed. This is due to the teacher model's inadequate learning of expressive features, which hampers the knowledge distillation process for the student model.

However, upon the removal of $\mathcal{L}_{mbm}$, the variant exhibits improved performance on FDA/GDB17. We conjecture that while the exclusive presence of $\mathcal{L}_{mam}$ might lead to features that favor a particular type of negative samples, such as molecules in GDB17 with fewer than 17 heavy atoms. However, the addition of $\mathcal{L}_{mbm}$ allows the model to be more generalizable to diverse datasets.

Results in line 4 indicates a performance drop. Counterintuitively, its performance is comparable or better than that of *w/o* $\mathcal{L}_{mam}$, aligning with the previous study (Ma et al., 2022). This is because training only with $\mathcal{L}_{mbm}$ is too simple due to the limited and imbalanced bond types, which results in a loss of specificity in the molecule features learned by the model. Even if the student model is distilled only on drug data, it can easily generalize to non-drug-like molecules, leading to poor performance.

**The effect of distillation losses.** From Table 2, we observe that employing any single distance-based loss to train the student model significantly outperforms the RNN model. Moreover, the

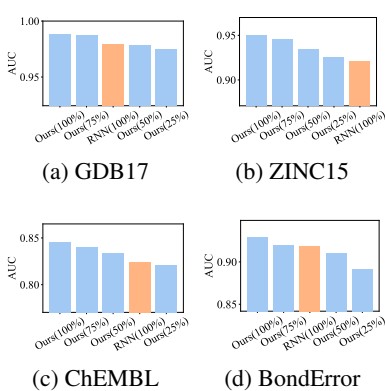

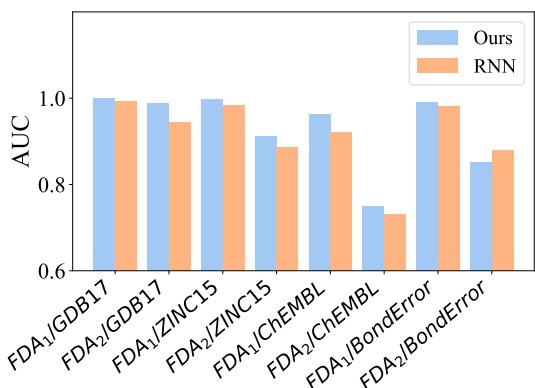

Figure 6: AUC values of our model and RNN using different amounts of training data. $Ours_r$ denotes our model trained with $r\%$ of training data.

Figure 7: The AUC values of RNN and our model in different test sets, where the FDA molecules are divided into two subsets based on the similarity to the training data.

simultaneous use of them enables our method to demonstrate optimal or near-optimal results across various test sets.

## 5.6 THE EFFECT OF TRAINING SET SIZE

In this subsection, we study the performance of our method w.r.t. the amount of training data in the knowledge distillation stage. We keep the teacher model unchanged, and use 25%, 50%, and 75% of original training samples to train several student models, respectively. Experimental results on different test sets are shown in Figure 6. Here, we also report the performance of the RNN model, which is the most competitive baseline according to the previously-reported results. It is very impressive that our model is still able to outperform RNN on the four test sets with only 50% training samples. In particular, on ZINC15, our method always surpasses the RNN model even with only 25% of the training data.

## 5.7 GENERALIZABILITY

To investigate the generalizability of our model, we calculate the maximum similarity between FDA molecules in the test sets and those in the training set. Utilizing a threshold of 0.5, we split the positive molecules in test sets into two subsets, $FDA_1$ and $FDA_2$, while retaining the negative molecules unchanged. These newly constructed test sets are employed to evaluate both the RNN model and our method.

As shown the Figure 7, from the test sets where $FDA_1$ molecules serve as positive molecules, it is evident that our method consistently outperforms the RNN model, demonstrating its proficiency in learning the distribution of drug molecules. Similarly, on the test sets where $FDA_2$ molecules are considered as positive molecules, our method exhibits superior performance in most sets. These findings suggest that in terms of generalization, our method surpasses the unsupervised drug-likeness prediction SOTA model.

## 6 CONCLUSION AND FUTURE WORK

In this paper, we make the first attempt towards unsupervised drug-likeness prediction based on 2D molecular structures, and propose a novel unsupervised method based on knowledge distillation. Concretely, we design two pre-training tasks: masked atom modeling and masked bond modeling, to train a teacher model, which can capture the topological features of molecules. Then, a student model is trained only on real drugs, inherently limiting its capability to mimic the teacher model's output when encounter with non-drug molecules. Finally, the gap between the outputs of the two models serves as an indicator of drug-likeness. In the future, we intend to explore how to enhance the parametric efficiency, with the aim of obtaining better performance with a smaller model.

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

# A    ATOM AND BOND FEATURES

As described in Section **??**, we introduce some predefined features for the initialization of atom and bond representations, which are as follow:

Table 3: Details of atom and bond features in molecular graphs

| Feature | Details |
|---|---|
| **Atom** | |
| Atom type | [1, 119] |
| Chirality tag | unspecifed, tetrahedral cw, tetrahedral ccw, other |
| Degree | [0, 10] |
| Formal charge | [-5, 5] |
| Number of H atoms | [0, 8] |
| Number of radicals | [0, 4] |
| Hybridization | sp, $sp^2$, $sp^3$, $sp^3d$, or $sp^3d^2$ |
| Atomaticity | 0 or 1 (aromatic atom or not) |
| Ring | 0 or 1 (atom is in a ring or not) |
| **Bond** | |
| Bond type | single, double, triple, aromatic |
| Conjugation | 0 or 1 (conjugated bond or not) |
| Stereochemistry | -, any, Z, E, cis, trans, |

# B    UPDATE OF ATOM PAIR REPRESENTATION

Following the previous studies (Jumper et al., 2021; Zhou et al., 2023; Lu et al., 2023a), the atom representation is updated by $\mathrm{Attention}(\cdot)$ with a bias term $\mathbf{B}$. Specifically, the atom representation $\boldsymbol{x}_i^{(l)}$ is updated as follows:

$$\boldsymbol{x}_i^{(l)} = \boldsymbol{x}_i^{(l-1)} + \mathrm{Attention}\left(\boldsymbol{x}_i^{(l-1)}, \boldsymbol{x}_j^{(l-1)}, \boldsymbol{p}_{ij}^{(l-1)}\right),$$
$$\boldsymbol{x}^{(l)} = \boldsymbol{x}^{(l)} + \mathrm{FFN}\left(\boldsymbol{x}^{(l)}\right). \tag{8}$$

Here, $\mathrm{FFN}(\cdot)$ is a one-layer feed forward network, and the $\mathrm{Attention}(\cdot)$ function is defined as

$$\mathrm{Attention}\left(\boldsymbol{x}_i^{(l-1)}, \boldsymbol{x}_j^{(l-1)}, \boldsymbol{p}_{ij}^{(l-1)}\right) =$$
$$\mathrm{softmax}\left(\frac{\boldsymbol{x}_i^{(l-1)}\mathbf{W}_Q^{(l,h)}\left(\boldsymbol{x}_j^{(l-1)}\mathbf{W}_K^{(l,h)}\right)^T}{\sqrt{d_h}} + \mathbf{B}_{i,j}^{(l,h)}\right)\boldsymbol{x}_j^{(l-1)}\mathbf{W}_V^{(l,h)}, \tag{9}$$

where $\mathbf{W}_Q^{(l,h)}$, $\mathbf{W}_K^{(l,h)}$, and $\mathbf{W}_V^{(l,h)}$ are trainable parameter matrices. The only difference between $\mathrm{Attention}(\cdot)$ and the standard self-attention in Transformer is the attention bias term:

$$\mathbf{B}_{i,j}^{(l,h)} = \boldsymbol{p}_{i,j}^{(l-1)}\mathbf{W}_B^{(l,h)}, \tag{10}$$

where the $\mathbf{W}_B^{(l,h)} \in \mathbb{R}^{d_p \times 1}$ is a trainable parameter matrix.

Meanwhile, The atom pair representation $\boldsymbol{p}_{i,j}^l$ is sequentially updated via $\mathrm{OuterProduct}(\cdot)$, $\mathrm{TrianglarUpdate}(\cdot)$ and $\mathrm{FFN}(\cdot)$, which is a a feed-forward network. The formulas are denoted as follows:

$$\boldsymbol{p}_{i,j}^{(l)} = \mathbf{P}^{(l-1)} + \mathrm{OuterProduct}\left(\boldsymbol{x}^{(l)}\right),$$
$$\boldsymbol{p}_{i,j}^{(l)} = \boldsymbol{p}_{i,j}^{(l)} + \mathrm{TriangularUpdate}\left(\boldsymbol{p}_{i,j}^{(l)}\right), \tag{11}$$
$$\boldsymbol{p}_{i,j}^{(l)} = \boldsymbol{p}_{i,j}^{(l)} + \mathrm{FFN}\left(\boldsymbol{p}_{i,j}^{(l)}\right),$$

The $\mathrm{OuterProduct}(\cdot)$ is used for atom-to-pair communication, which is denoted as:

$$\boldsymbol{a} = \boldsymbol{x}^{(l)}\mathbf{W}_{O1}^{(l)}, \boldsymbol{b} = \boldsymbol{x}^{(l)}\mathbf{W}_{O2}^{(l)}$$
$$\boldsymbol{o}_{i,j} = \text{flatten}\left(\boldsymbol{a}_i \otimes \mathbf{b}_j\right); \tag{12}$$
$$\text{output} = \boldsymbol{o}\mathbf{W}_{O3}^{(l)},$$

where $\mathbf{W}_{O1}^{(l)} \in \mathbb{R}^{d_a \times d_o}$, $\mathbf{W}_{O2}^{(l)} \in \mathbb{R}^{d_a \times d_o}$ and $\mathbf{W}_{O3}^{(l)} \in \mathbb{R}^{d_o^2 \times d_p}$ are trainable parameters. $d_o$ is the dimension of the $\text{OuterProduct}(\cdot)$. $\boldsymbol{a}$, $\boldsymbol{b}$ and $\boldsymbol{o}$ are temporary variables in this function, and $\boldsymbol{o} = [\boldsymbol{o}_{i,j}]$. Furthermore, we use $\text{TrianglarUpdate}(\cdot)$ to enhance the atom pair representations, denoted as:

$$\boldsymbol{a} = \text{sigmoid}\left(\boldsymbol{p}^{(l)}\mathbf{W}_{T1}^{(l)}\right) \odot \left(\boldsymbol{p}^{(l)}\mathbf{W}_{T2}^{(l)}\right);$$
$$\boldsymbol{b} = \text{sigmoid}\left(\boldsymbol{p}^{(l)}\mathbf{W}_{T3}^{(l)}\right) \odot \left(\boldsymbol{p}^{(l)}\mathbf{W}_{T4}^{(l)}\right);$$
$$\boldsymbol{o}_{i,j} = \sum_k \boldsymbol{a}_{i,k} \odot \boldsymbol{b}_{j,k} + \sum_k \boldsymbol{a}_{k,i} \odot \boldsymbol{b}_{k,j}; \tag{13}$$
$$\text{output} = \text{sigmoid}\left(\boldsymbol{p}^{(l)}\mathbf{W}_{T5}^{(l)}\right) \odot \left(\boldsymbol{o}\mathbf{W}_{T6}^{(l)}\right),$$

where $\mathbf{W}_{T1}^{(l)}, \mathbf{W}_{T2}^{(l)}, \mathbf{W}_{T3}^{(l)}, \mathbf{W}_{T4}^{(l)} \in \mathbb{R}^{d_p \times d_t}$, $\mathbf{W}_{T5}^{(l)} \in \mathbb{R}^{d_p \times d_p}$, and $\mathbf{W}_{T6}^{(l)} \in \mathbb{R}^{d_t \times d_p}$ are trainable parameters. $d_t$ is the dimension of the $\text{TrianglarUpdate}(\cdot)$.

## C   DETAILS OF DATASET

To investigate the effectiveness of our method, we conduct several groups of experiments on the commonly-used datasets. We employ the small molecules from the ChEMBL database to pre-train our teacher model due to its extensive coverage and demonstrated bioactivity against proteins. Following the settings in Lee et al. (2022), we utilize the the Worlddrug dataset (approved drugs) as the training data for our student model.

Molecules from the FDA dataset serve as positive samples in the test set, the number of which is up to 1,489, and the negative samples are from various databases: GDB17 (Ruddigkeit et al., 2012), ZINC15 (Sterling & Irwin, 2015), and ChEMBL. Recently, molecule generation models based on deep learning (Shi et al., 2020; Kuznetsov & Polykovskiy, 2021; Adams & Coley, 2023; Lu et al., 2023b) have numerous applications in drug discovery. However, they often tend to generate molecules with unreasonable bonds that are difficult to be filtered out by existing methods, posing a primary challenge for drug-likeness prediction models in practical applications. To examine whether our model can address this issue, we construct a new test set, termed BondError. It is created by initially selecting high-quality molecules that satisfy predefined rules from the ChEMBL database. Subsequently, we randomly substitute double bonds with single bonds in these molecules. The altered molecules must not only differ from their original counterparts but also be identifiable by RDKit.

## D   DIFFERENT SIMILARITY THRESHOLDS

To evaluate the generalizability our model, we use lower similarity thresholds to partition the training set, which increases the difficulty of model inference. Specifically, the similarity thresholds range from 0.6 to 0.8, with an interval of 0.05. The results in Table 5 demonstrate that our model consistently outperforms RNN across all threshold settings.

Table 4: Detailed description of all datasets

|  | Data name | Composition | Description |
|---|---|---|---|
| Training | Worlddrug | 2833 Worlddrug molecules | Drug dataset for training unsupervised baselines and our model |
|  | Worlddrug/ZINC15 | 2833 Worlddrug molecules as positive samples and 2833 ZINC15 molecules as negative samples | Dataset for training the GCN-based supervised model |
| Test | FDA/GDB17 | 1489 FDA molecules as positive samples and 10,000 GDB17 molecules as negative samples | GDB17 molecules are generated by a graph enumeration method, which are highly non-drug-like |
|  | FDA/ZINC15 | 1489 FDA molecules as positive samples and 10,000 ZINC15 molecules as negative samples | The chemical space of ZINC15 dataset is more drug-like than GDB17 |
|  | FDA/ChEMBL | 1489 FDA molecules as positive samples and 10,000 ChEMBL molecules as negative samples | The molecules from ChEMBL are with a pChEMBL value of 5.85 or higher and thus are more drug-like than ZINC15 |
|  | FDA/BondError | 1489 FDA molecules as positive samples and 10,000 molecules with bond-level errors as negative samples | The molecules in BondError are from the ChEMBL database, and some double bonds are randomly replaced with single bonds, which are different from the original molecules but keep the RDKit readable |

Table 5: Results of RNN and ours based on different similarity thresholds.

|  | FDA/GDB17 | | FDA/ZINC15 | | FDA/ChEMBL | | FDA/BondError | |
|---|---|---|---|---|---|---|---|---|
|  | Ours | RNN | Ours | RNN | Ours | RNN | Ours | RNN |
| 0.8 | 0.991±0.000 | 0.966±0.001 | 0.948±0.001 | 0.927±0.001 | 0.840±0.001 | 0.812±0.001 | 0.927±0.002 | 0.924±0.001 |
| 0.75 | 0.991±0.000 | 0.966±0.001 | 0.948±0.001 | 0.927±0.001 | 0.840±0.001 | 0.812±0.001 | 0.927±0.002 | 0.924±0.001 |
| 0.7 | 0.993±0.001 | 0.966±0.001 | 0.948±0.002 | 0.928±0.001 | 0.840±0.000 | 0.811±0.001 | 0.929±0.001 | 0.923±0.000 |
| 0.65 | 0.992±0.001 | 0.964±0.000 | 0.949±0.000 | 0.924±0.001 | 0.834±0.001 | 0.803±0.000 | 0.927±0.001 | 0.918±0.001 |
| 0.6 | 0.992±0.001 | 0.964±0.001 | 0.940±0.001 | 0.921±0.002 | 0.821±0.001 | 0.795±0.005 | 0.919±0.003 | 0.915±0.004 |

