# OpenReview forum: "Unsupervised 2D Molecule Drug-likeness Prediction based on Knowledge Distillation"
_ICLR.cc/2025/Conference — ICLR 2025 Conference Withdrawn Submission_

### Official Review · Reviewer_v5s9 · 2024-10-31

**Soundness:** 2
**Presentation:** 3
**Contribution:** 2
**Rating:** 3
**Confidence:** 1

**Summary:**

The paper addresses drug-likeness prediction challenges and introduces a novel knowledge distillation approach. In this method, a teacher model is pretrained using 2D molecular graphs with atom/bond masking predictive modeling, trained on a large dataset comprising both drugs and non-drugs. The student model, by contrast, is trained solely on drugs, separate from the teacher's dataset. The final drug-likeness prediction is based on the difference in likelihood predictions between the teacher and student models.

The authors evaluate their method using standard benchmarks, comparing it to five baselines. The baselines include two classes: supervised approaches (QED, a graph neural network (GCN), and a recurrent neural network (RNN)) and unsupervised methods (GlocalKD and HimNet). In four subsets of the FDA-approved drugs dataset, the proposed approach significantly outperforms these baselines. An ablation study is also conducted to examine the contributions of pretraining and distillation, alongside analyses highlighting the RNN’s bias toward shorter drug molecules. While the source code is not yet available, the authors have committed to open-sourcing it in the future.

**Strengths:**

The paper was well written, the related works and the motivation behind their methods is well explained.
The idea of using the difference between the teacher models and student models likeliness prediction is interesting.
Interesting analysis on the bias of RNN toward short sequences.

**Weaknesses:**

Although this work targets a specific molecular property prediction task, it does not thoroughly discuss or compare against a substantial body of research in molecular representation learning. For instance, methods based on molecular fingerprints and GNNs, such as *ADMET Property Prediction through Combinations of Molecular Fingerprints* ([arXiv:2310.00174](https://arxiv.org/abs/2310.00174)), have shown strong results in ADMET prediction and could be readily adapted to tasks like drug-likeness prediction. Additionally, recent advancements in pretrained models—such as *Molformer* ([Nature](https://www.nature.com/articles/s42256-022-00580-7)), *Graphormer* ([GitHub](https://github.com/microsoft/Graphormer)), and *ImageMol* ([GitHub](https://github.com/HongxinXiang/ImageMol))—would be valuable baseline comparisons for the present study.

The novelty of the proposed pretraining tasks also appears limited, as atom and bond masking in graph pretraining has become a widely adopted approach. For example, *GraphMVP* ([OpenReview](https://openreview.net/pdf?id=xQUe1pOKPam)) employs similar masking strategies to pretrain GNNs, covering both 2D and 3D graphs, with masking applied to parts of 2D graphs as well.

Furthermore, the source code has not been made publicly available, hindering reproducibility of the experimental results.

**Questions:**

Could you please consider a comparison with the baselines Molformer, Graphformer and ImageMol when they are finetuned on the druglikeliness prediction tasks?

Could you please compare to the GraphMVP methods?

---

### Official Review · Reviewer_YU7Z · 2024-11-02

**Soundness:** 2
**Presentation:** 3
**Contribution:** 2
**Rating:** 5
**Confidence:** 4

**Summary:**

The authors proposed a 2D-based unsupervised drug-likeness prediciont method. They performed knowledge distribution by pretraining a teacher model on both positive and negative molecules and futher trained a student model on positive drug-like molecules only, and further minimized the embedding between teacher model and student model.

**Strengths:**

The paper is well written and easy to follow.

**Weaknesses:**

There are some major concerns about this paper:
1. The performance of the RNN paper looks great enough according to Table 1, even in the BondError dataset proposed by the authors, the RNN performance is pretty great. I see the main disadvantage of the RNN method is about its bias on the SMILES length. Thus, the authors should proposed some new datasets which contain molecules with different scales of SMILES lengths. Even though the authors showed the comparsion between RNN and their method on different scales of SMILES lengths in Figure 5, which partially address this concern, it's still not that complete. And the authors didn't display the number of molecules for different lengths.
2. The baseline methods are too weak. The RNN method was way too old. Even the SMILES-BERT was trained 5 years ago. I wonder if the authors would use any transformer for comparison.

There are some other minor concerns about this paper:
1. The score is based on atom embedding level, why not consider bond embedding as well?
2. Be more careful about the potential data leakage problem, even though it might be hard to avoid when there is pretraining stage. Consider scaffold split.
3. There are two $L_{mam}$ in Formula (1)
4. Only positive examples are used in the training of the student model, what if introduce some negative examples and perform constrastive learning?

**Questions:**

See weaknesses

---

### Official Review · Reviewer_aPdR · 2024-11-02

**Soundness:** 2
**Presentation:** 3
**Contribution:** 2
**Rating:** 3
**Confidence:** 4

**Summary:**

This paper focuses on drug-likeness prediction based on chemical structures. Instead of framing the problem as a supervised task or likelihood-based estimation, this work proposes an approach based on self-supervised learning followed by knowledge distillation. The proposed method  ios compared against several baselines, including supervised classification, likelihood-based (RNN), and QED. The approach is tested on multiple datasets. Multiple analysis and ablation studies are conducted, providing further insights into the results.

**Strengths:**

- The topic of the paper is relevant, as an improved quantification of drug-likeness can accelerate the drug discovery process and enable other approaches.
- The method is clearly explained.
- Analysis and ablation studies help develop an understanding of the proposed approach.

**Weaknesses:**

The main limitations of this work are related to its novelty,  lack of baselines, and limited clarity on its overall positioning.
- First of all, in the introduction and motivation, the paper distinguishes itself from other unsupervised-based approaches based on the fact that previous work leverages SMILES representations, while this work leverages 2D graphs. However, it is actually possible (and typically done) to compute likelihoods based on 2D graph representations. Indeed, this is typically one of the main ways graph (and molecule) generative methods are evaluated (see, e.g., Diamant et al., 2023 ICML). It is in general well known that for molecular tasks, graph-based representations outperform SMILES-based representations, both for supervised and generative tasks (see, for example, leaderboard https://ogb.stanford.edu/docs/leader_graphprop/). Therefore, using graph-based representations (which have been state-of-the-art for years) instead of SMILES-based representations does not seem to be particularly novel.
- This paper introduced a self-supervised framework that appears to be very similar to previous work (see "Evaluating Self-Supervised Learning for Molecular Graph Embeddings", NeurIPS 2023 for some examples). In this context, the choice of the self-supervised model introduced in this paper appears not novel and arbitrary.
- This method is framed as novel compared to existing methods based on outlier-based estimation. However, the proposed approach is actually an outlier estimation technique, given that the drug-likeness score is obtained as difference between a model trained on the whole chemical space, and a model trained only on drug-like (i.e., "known") molecules. Therefore, more advanced outlier estimation methods should be evaluated.

Overall, it is not clear what the contribution and novelty of this paper is. Additionally, several critical baselines are missing.

**Questions:**

- The authors should better clarify what the novelty of this work is, also accounting for the comments above.
- The authors should introduce more baselines, in particular focusing on:
    - State-of-the-art molecular generative methods (e.g., based on graph-based representations) used to estimate likelihoods, instead of SMILES-based.
    - Other self-supervised methods used to learn general chemical representations, and to define the chemical space.
    - Outlier detection methods used to define novelty.

In this context, the authors should better clarify the original contributions proposed by this work.

---

### Official Review · Reviewer_V4ux · 2024-11-03

**Soundness:** 2
**Presentation:** 3
**Contribution:** 2
**Rating:** 5
**Confidence:** 4

**Summary:**

The paper presents an unsupervised method for predicting drug-likeness in molecules that exploits 2D features of molecules. It uses a knowledge distillation approach with two models: a "teacher" model trained on a large dataset of molecules, which learns molecular topology through tasks like masked atom and bond prediction, and a "student" model trained only on real drugs. The student model mimics the teacher’s output on drug-like molecules but diverges on non-drug molecules, allowing for a drug-likeness score based on the difference between the models’ outputs. Experimental results show that this method outperforms existing models and is less affected by biases, offering a potentially more accurate way to determine drug likeliness.

**Strengths:**

1. The paper offers a scalable approach for drug-likeness screening, with practical applications in drug discovery and unsupervised molecular learning.
2. By using 2D molecular graphs instead of SMILES, the approach effectively reduces biases commonly associated with SMILES-based drug-likeness scoring.
3. The method consistently demonstrates superior performance compared to baseline models, highlighting its robustness and effectiveness.
4. The knowledge distillation approach proposed in the paper might be an effective way to address challenges with unbalanced datasets in drug discovery, where true positives are often limited.

**Weaknesses:**

1. The scoring method relies solely on the difference between teacher and student models. Including additional criteria, such as molecule toxicity features, could improve robustness.
2. While the model leverages 2D molecular graphs, drug effectiveness often depends on 3D molecular interactions with proteins, which this paper does not address as a limitation.
3. To assess the model's true potential in drug discovery, testing on novel, unseen datasets and conducting out-of-distribution benchmarks would be valuable.
4. In practical applications like drug discovery, an interpretability analysis would be beneficial to understand the model’s behavior.

**Questions:**

1. Interpretability of Scoring: Could the authors clarify how the gap between teacher and student outputs specifically reflects drug-likeness, possibly by linking it to characteristics like toxicity markers or functional groups?
2. Hyperparameter Sensitivity: How sensitive is the model to masking ratios in atom/bond modeling tasks?

---

### Note · Authors · 2025-01-16

I have read and agree with the venue's withdrawal policy on behalf of myself and my co-authors.